# *CodeMirage*: STRESS-TESTING AI-GENERATED CODE DETECTORS AGAINST PRODUCTION-LEVEL LLMS

## ABSTRACT

Large language models (LLMs) are increasingly integrated into software development, generating substantial volumes of source code. While they enhance productivity, their misuse raises serious concerns, including plagiarism, license violations, and the propagation of insecure code. Robust detection of AI-generated code is therefore essential, and requires benchmarks that faithfully reflect real-world conditions. Existing benchmarks, however, are limited in scope, covering few programming languages and relying on less capable models. In this paper, we introduce *CodeMirage*, a comprehensive benchmark that addresses these gaps through three key contributions: (1) coverage of ten widely used programming languages, (2) inclusion of both original and perturbed code from ten state-of-the-art, production-level LLMs, and (3) six progressively challenging tasks across four evaluation configurations. Using *CodeMirage*, we evaluate ten representative detectors spanning four methodological paradigms under realistic settings, with performance reported across three complementary metrics. Our analysis yields eight key findings that reveal the strengths and limitations of current detectors and highlight critical challenges for future research. We believe *CodeMirage* provides a rigorous and practical testbed to drive the development of more robust and generalizable AI-generated code detectors.

## 1 INTRODUCTION

Large Language Models (LLMs) are rapidly evolving and demonstrating increasing capabilities in coding, fundamentally transforming the software development ecosystem. Recent LLMs such as ChatGPT (OpenAI, 2022) and Claude (Anthropic, 2025) exhibit remarkable code generation performance, producing high-quality outputs in response to concise natural language prompts. The emergence of reasoning-capable models like DeepSeek-R1 (Guo et al., 2025) has further accelerated LLM adoption among developers. According to Stack Overflow's industry report (Stack Overflow, 2024), 82.1% of the 65,000 surveyed developers report using ChatGPT (OpenAI, 2022) during their development workflow. Capitalizing on the strong coding abilities of LLMs, assistant tools such as GitHub Copilot (Friedman, Nat, 2022) and Cursor (Cursor, 2023) have been developed to enhance productivity by helping developers write, modify, and debug code directly within integrated development environments (IDEs). Furthermore, state-of-the-art LLM-based agentic systems such as OpenHands (Wang et al., 2025) achieve up to a 65.8% resolved rate on SWE-Bench (Jimenez et al., 2024), demonstrating the effectiveness of LLMs in addressing real-world software engineering tasks. These trends indicate that LLMs and their associated tools are becoming integral to modern software development workflows.

However, the rapid spread of AI-generated code has raised concerns about new vulnerabilities and misuse. Systematic benchmarks show that LLM outputs often ship with logic errors and latent security flaws (Liu et al., 2023; Gao et al., 2023; Tóth et al., 2024; Zhang et al., 2023; Pearce et al., 2022; Kaniewski et al., 2024). Comparative evaluations reveal that AI suggestions can embed at least as many vulnerabilities as human code (Khoury et al., 2023; Wang et al., 2024; Tambon et al., 2025; Vaidya & Asif, 2023; Asare et al., 2023; Tihanyi et al., 2025). Furthermore, LLMs are susceptible to manipulation (Kaniewski et al., 2024), including poisoning attacks (Yan et al., 2024; Cotroneo et al., 2024; Oh et al., 2024) and prompt injections (Mastropaolo et al., 2023; Zeng et al., 2025), which can induce the generation of targeted vulnerable code. At the same time, educators warn of an impending wave of AI-driven plagiarism that evades conventional detectors (Hutson,

2024; Steponenaite & Barakat, 2023; Khalaf, 2025; Dehouche, 2021; Xiao et al., 2022; Sağlam et al., 2024; Khalil & Er, 2023), while legal scholars highlight intellectual-property (Yu et al., 2023; Li et al., 2023; Xu et al., 2024a; Stalnaker et al., 2024) and licence-compliance (Xu et al., 2024c) risks. Robust AI-code detection is therefore critical for secure software supply chains, responsible academic practice, and licence compliance.

To address the challenges of AI-generated code identification, various detection methods have been proposed, leveraging statistical features of code (Idialu et al., 2024), the capabilities of language models (Xu & Sheng, 2024; Shi et al., 2025; Yang et al., 2023; Ye et al., 2025; Xu et al., 2025; Nguyen et al., 2024; 2023), and code embedding models (Suh et al., 2025; Liu et al., 2024b). However, evaluations based on existing benchmarks and datasets (Suh et al., 2025; Pan et al., 2024; Demirok & Kutlu, 2024; Pham et al., 2024; Orel et al., 2025; Xu et al., 2024b) often fall short in three key aspects. First, they typically cover only a narrow set of programming languages—primarily C++ and Python—while neglecting other widely used languages such as Go and HTML, resulting in limited language diversity compared to real-world software development. Second, most benchmarks rely on open-source LLMs with relatively small model sizes and lower generation quality, or include only a small number of commercial models, leaving a gap between benchmark conditions and real-world usage. Third, most existing datasets lack practical adversarial scenarios, such as paraphrasing (Krishna et al., 2023; Sadasivan et al., 2023), which are common in practice and essential for evaluating the robustness of detection systems. Thus, a rigorous benchmark that captures real-world language diversity, modern commercial models, and adversarial scenarios is indispensable for driving meaningful progress in this emerging field.

We hence introduce *CodeMirage*, a comprehensive benchmark for stress-testing AI-generated code detectors under realistic and adversarial conditions, to solve the three major limitations identified in prior benchmark work. *CodeMirage* is constructed from real-world human-written code and enriched with both AI-generated and perturbed variants produced by a diverse set of state-of-the-art reasoning and non-reasoning LLMs from six major commercial service providers.

Our key contributions are as follows:

- We present a large-scale, multilingual benchmark for stress-testing AI-generated code detection, spanning 10 widely used programming languages. The dataset comprises approximately 210,000 samples, including 10,000 human-written code files sourced from GitHub (CodeParrot, 2022b), as well as AI-generated and perturbed counterparts produced by 10 production-level LLMs.
- We design four progressively challenging evaluation configurations with six tasks and three complementary performance metrics to facilitate rigorous and realistic assessment of detector effectiveness under various real-world scenarios.
- We conduct a comprehensive evaluation of 10 representative detectors across four methodological paradigms using *CodeMirage*, providing insights into their accuracy, robustness, and generalization across program languages, models, and adversarial settings.

## 2 BACKGROUND AND RELATED WORK

### 2.1 TAXONOMY OF AI-GENERATED CODE DETECTORS

Detecting AI-generated content has been a long-standing challenge in both the natural language (Uchendu et al., 2021; Gehrmann et al., 2019; Akram, 2023; Ghosal et al., 2023) and computer vision domains (Rössler et al., 2018; Güera & Delp, 2018; Zhu et al., 2023; Dolhansky et al., 2020; Zi et al., 2020), predating even the emergence of large language models (LLMs) (Vaswani et al., 2017; Achiam et al., 2023) and diffusion-based generative models (Sohl-Dickstein et al., 2015; Ho et al., 2020). In contrast, detecting AI-generated source code is a relatively new research direction, emerging primarily in the last two years due to the rapid advancements in the coding capabilities of LLMs (OpenAI, 2022; Anthropic, 2025).

Inspired by traditional statistical-based methods used for AI-generated text detection (Ramos et al., 2003; Ippolito et al., 2020), early approaches for code focus on analyzing surface-level statistical features. For example, Whodunit (Idialu et al., 2024) extracts stylometric and complexity-based features from both raw source code and its abstract syntax tree (AST). However, these methods

Table 1: Comparison between existing AI-generated code benchmarks and our *CodeMirage*. **Gran.** = granularity (*Func*: function/snippet, *Doc*: whole file). IID = in–distribution; OOD = out-of-distribution. Baseline categories: **Z** (zero-shot detector), **E** (embedding-based detector), **F** (fine-tuning-based detector), **P** (pre-trained LLM + downstream detector). Columns "Open LLMs" and "Comm. LLMs" show whether the dataset includes *any* open-source or commercial generators.

| Dataset ↓ Stat.→ | #Lang | Gran. | IID | OOD | #Open LLMs | #Comm. LLMs | Reasoning Model | #Human Code | #AI Code | Adv. Test | Quality Check | Baseline #/Cat. |
|---|---|---|---|---|---|---|---|---|---|---|---|---|
| Suh *et al.* (Suh et al., 2025) | 3 | Func | ✓ | ✓ | 1 | 3 | ✗ | $\sim 3.7k$ | $\sim 29.5k$ | ✗ | ✗ | 8 / Z,E,F |
| Pan *et al.* (Pan et al., 2024) | 1 | Func | ✓ | ✓ | 0 | 1 | ✗ | $\sim 5k$ | $\sim 71k$ | ✓ | ✗ | 5 / Z |
| AIGCodeSet (Demirok & Kutlu, 2024) | 1 | Func | ✓ | ✗ | 2 | 1 | ✗ | $\sim 4.8k$ | $\sim 2.9k$ | ✗ | ✓ | 3 / E,F |
| MAGECODE (Pham et al., 2024) | 3 | Doc | ✓ | ✗ | 0 | 3 | ✗ | $\sim 81k$ | $\sim 45k$ | ✗ | ✓ | 8 / Z |
| CoDet-M4 (Orel et al., 2025) | 3 | Func | ✓ | ✓ | 4 | 1 | ✗ | $\sim 252k$ | $\sim 246k$ | ✓ | ✓ | 6 / F,P |
| LLMGCode (Xu et al., 2024b) | 8 | Doc | ✓ | ✗ | 1 | 3 | ✗ | $< 1k$ | $2k$ | ✗ | ✗ | 10 / Z,F,P |
| *CodeMirage (Ours)* | 10 | Doc | ✓ | ✓ | 4 | 6 | ✓ | $10k$ | $\sim 200k$ | ✓ | ✓ | 10 / Z,E,F,P |

often struggle to distinguish code generated by modern, high-performing LLMs (OpenAI, 2022; Anthropic, 2025; Guo et al., 2025; Kavukcuoglu, 2025), which better mimic human coding styles.

To improve detection effectiveness, recent research has explored more advanced techniques — often leveraging large language models (LLMs) or code embedding models — which can be broadly categorized into the following four methodological paradigms:

**Zero-shot Detector.** These detectors assign confidence scores from token-level statistics of pretrained LLMs without task-specific fine-tuning. LogRank (Gehrmann et al., 2019) and Entropy (Lavergne et al., 2008) use average next-token log-rank and entropy to capture AI-generated code distributions. DetectGPT (Mitchell et al., 2023) measures divergence between original and perturbed text, a strategy extended to code by DetectCodeGPT (Shi et al., 2025), GPT4Code (Yang et al., 2023), and AIGC Detector (Xu & Sheng, 2024) with code-specific perturbations. CR (Ye et al., 2025) instead compares original and LLM-rewritten code, while Binoculars (Hans et al., 2024) employs cross-perplexity between instruction-tuned and non-instruction-tuned LLMs.

**Embedding-based Detector.** Embedding-based detectors (Khoury et al., 2023) utilize pretrained code embedding models, such as CodeT5+ Embedding (Wang et al., 2023) and CodeXEmbed (Liu et al., 2024b), to extract high-level semantic representations from either raw source code or abstract syntax trees (ASTs). These embeddings are then fed into lightweight classifiers, *e.g.*, MLP (Rosenblatt, 1958), to perform binary classification between human-written and AI-generated code.

**Fine-tuning-based Detector.** This class of detectors fine-tunes transformer-based models to directly capture discriminative patterns between human-written and AI-generated code. For example, GPTSniffer (Nguyen et al., 2023; 2024) fine-tunes CodeBERT (Feng et al., 2020) on labeled code samples to perform binary classification. Other approaches (Suh et al., 2025) explore different backbone architectures, such as CodeT5+ (Wang et al., 2023) and RoBERTa (Liu et al., 2019).

**Pretrained LLM with Downstream Detector.** These methods extract semantic or statistical features from pretrained LLMs and train downstream classifiers on them. MageCode (Pham et al., 2024) leverages hidden-state statistics from the classification token of CodeT5+ (Wang et al., 2023) with a two-layer linear classifier. Raidar (Mao et al., 2024), originally for text, can be adapted to code by comparing metrics between original and LLM-rewritten samples and training an XGBoost (Chen et al., 2025) classifier. BiScope (Guo et al., 2024) introduces a bi-directional cross-entropy analysis using pretrained LLMs, with features classified via Random Forest (Breiman, 2001).

## 2.2 Existing AI-generated Code Datasets and Benchmarks

Prior studies (Suh et al., 2025; Pan et al., 2024; Demirok & Kutlu, 2024; Pham et al., 2024; Orel et al., 2025; Xu et al., 2024b) has laid important groundwork for building benchmarks to evaluate AI-generated code detectors. As shown in Table 1, several benchmarks introduce valuable contributions: for instance, Suh *et al.* (Suh et al., 2025) propose a large-scale function-level dataset spanning three programming languages. Pan *et al.* (Pan et al., 2024) and CoDet-M4 (Orel et al., 2025) incorporate adversarial perturbations into AI-generated code to test robustness. AIGCodeSet (Demirok & Kutlu, 2024) and MAGECODE (Pham et al., 2024) employ quality checks during code generation. LLMGCode (Xu et al., 2024b) expands language coverage to eight programming languages. Collectively, these datasets serve as solid foundations for evaluating AI-generated code detectors.

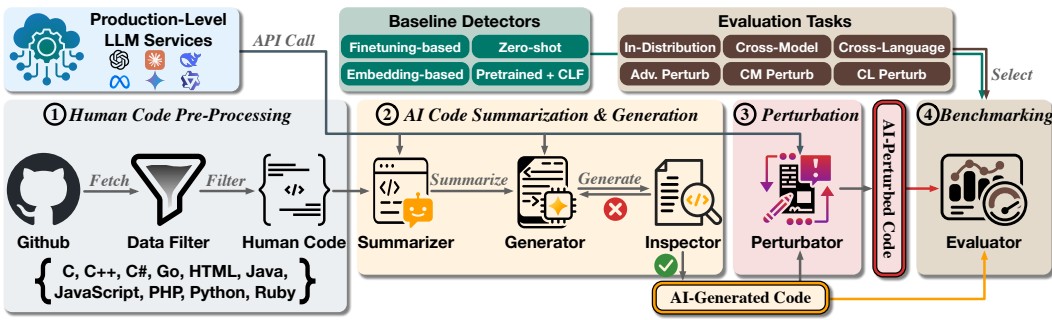

Figure 1: Overview of the ***CodeMirage*** framework. We collect and preprocess human-written code from GitHub repositories, then leverage 10 state-of-the-art LLMs to summarize, generate, and perturb code with quality inspection. Finally, ***CodeMirage*** evaluates 10 AI-generated code detectors across six evaluation tasks, covering a wide range of real-world scenarios.

However, these benchmarks have notable limitations. Most cover only a small number of programming languages, rely on less capable LLMs, and none of them leverage latest reasoning models (Guo et al., 2025; OpenAI, 2025; Kane, 2025). Furthermore, baseline evaluations in these benchmarks do not comprehensively include all four major categories of detection methods, and only two existing benchmarks include adversarial test, which is critical for real-world deployment.

To address these gaps, our proposed benchmark, ***CodeMirage***, includes: (1) code samples across 10 widely used programming languages; (2) outputs from 10 state-of-the-art production-level LLMs, including three reasoning models; (3) both out-of-distribution and adversarial evaluation settings; and (4) baselines covering all four methodological categories of AI-generated code detection.

## 3 ***CodeMirage***'S DATASET CONSTRUCTION

### 3.1 HUMAN CODE PRE-PROCESSING

To construct a comprehensive benchmark for AI-generated and paraphrased source code, we begin by sourcing high-quality human-written samples from the CodeParrot GitHub-Code-Clean dataset (CodeParrot, 2022b) — a curated subset of the original GitHub-Code corpus (CodeParrot, 2022a), as illustrated in Figure 1. This cleaned dataset excludes overly short snippets, auto-generated files, and code with excessive alphanumeric or non-semantic content. Importantly, the dataset was collected and sanitized in May 2022, before the widespread deployment of modern code-generating LLMs and AI-assisted coding agents. This timing ensures that the selected code examples are genuinely human-authored. Based on the dataset's language distribution, we select the ten most widely used programming languages, including C, C++, C#, Go, HTML, Java, JavaScript, PHP, Python, and Ruby. For each language, we randomly extract 1,000 representative code snippets. To preserve diversity while maintaining tractability, we apply additional length-based filtering, ensuring that the final samples span a broad range of real-world patterns but remain within a controlled length scale. Additional details regarding ethical considerations and mitigation of potential dataset bias during the human code pre-processing stage are provided in Appendix A.

### 3.2 AI CODE SUMMARIZATION AND GENERATION

Following the collection of human-written code, we employ ten state-of-the-art, production-level LLMs to generate their AI counterparts (detailed in Appendix B). The process begins with AI Code Summarization. To ensure high-quality generation and prevent direct copying, ***CodeMirage*** adopts a text-to-code strategy in which each human-written code sample is first summarized into a descriptive summary. The human samples in ***CodeMirage*** are often comprehensive code files, encompassing library imports, class and structure definitions, and detailed function logic. To distill this information, we prompt the LLMs to summarize the sample's *purpose*, *functionality*, *logic overview*, and *key features*. The summaries also include the names of relevant *libraries*, *functions*, *classes*, *structures*, and *variables*, along with any *optional contextual notes* describing uncommon assumptions or external dependencies. Full prompts and example summaries are provided in Appendix C.

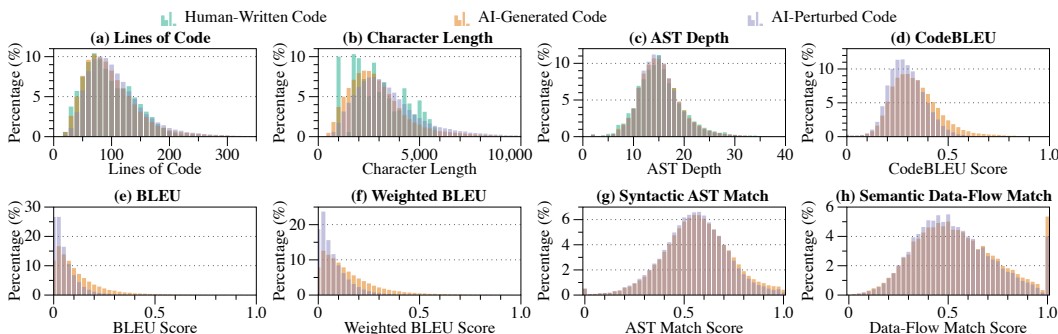

Figure 2: Benchmark statistics of *CodeMirage*.

These summaries act as intermediate representations and are subsequently used as input prompts in the next phase: AI Code Generation. Given the descriptive summary of each human-written code sample, *CodeMirage* employs LLMs to generate corresponding AI code. To better align structural characteristics with the original human sample, we additionally include metadata — such as target line count and character length — within the prompt. Due to the inherent stochasticity of LLMs, generated code may occasionally deviate from the desired format or exhibit insufficient novelty. To ensure quality and distinctiveness, we apply a rule-based inspector enforcing two key constraints: (1) consistency with the original sample's line count and character length, and (2) adequate token-level divergence from the original, enforced by requiring a BLEU (Papineni et al., 2002) score below 0.5 to avoid recitation (the rationale for this threshold is detailed in Appendix D). Samples failing either criterion are regenerated, with repeated failures resulting in exclusion. Detailed prompts and representative generation examples are included in Appendix E.

## 3.3 AI Code Perturbation

The primary perturbation technique employed in *CodeMirage* is Paraphrasing (Krishna et al., 2023; Sadasivan et al., 2023), a widely recognized method for evaluating the robustness of AI-generated text detectors under adversarial and real-world conditions. Despite its relevance, most existing benchmarks for AI-generated code detection (Suh et al., 2025; Pan et al., 2024; Demirok & Kutlu, 2024; Pham et al., 2024; Orel et al., 2025; Xu et al., 2024b) do not incorporate such adversarial testing. While a few studies in text detection (Mao et al., 2024; Guo et al., 2024) have included paraphrased code in their evaluations, they typically utilize generic prompting schemes and limited code samples, thereby restricting the scope and impact of their assessments. In *CodeMirage*, we introduce a systematic, domain-specific paraphrasing for code, covering six transformation types. These transformations are carefully designed to reflect realistic developer behaviors and common coding style variations. Detailed rules, prompt designs, and representative examples are provided in Appendix F. Additionally, we incorporate three advanced adversarial techniques — Multi-Round Paraphrasing, DeepWordBug (Gao et al., 2018), and AST-based Perturbation — to further stress-test detector robustness. While these advanced attacks are evaluated on a smaller subset of the dataset, they offer valuable insights into more targeted and extreme adversarial conditions. The design and evaluation of these attacks are detailed in Appendix G.

## 3.4 Benchmark Statistics

*CodeMirage* covers ten programming languages, each containing 1,000 human-written code samples with 10,000 AI-generated counterparts and 10,000 AI-perturbed counterparts. For every language, we obtain 2,000 outputs from each of ten production-level LLMs, yielding a 1:20 mapping between every human sample and its LLM-generated variants. Within every 1,000-sample shard (human or AI), we allocate 700 examples for training and 300 for testing.

We present four structural and semantic metrics of the dataset in Figure 2: lines of code (a), character length (b), AST depth (c), and CodeBLEU (Ren et al., 2020) score (d). The first three metrics reflect the overall structural characteristics of the code and show close resemblance between human-written and AI-generated samples. This similarity implies that naive statistical classifiers would struggle to detect AI-generated code using basic code features.

Figure 2 (d) reports the CodeBLEU score, a composite metric calculated as:

$$CodeBLEU = \alpha \cdot BLEU + \beta \cdot BLEU_{weighted} + \gamma \cdot Match_{AST} + \delta \cdot Match_{DF}, \quad (1)$$

where each component is equally weighted with $\alpha = \beta = \gamma = \delta = 0.25$ by default. The median CodeBLEU score for AI-generated code is approximately 0.3, consistent with prior observations in text-to-code generation (Dong et al., 2023; Espejel et al., 2023; Evtikhiev et al., 2023). Perturbed code yields slightly lower scores due to deliberate perturbations in both code format and structure.

To further analyze *CodeMirage*'s code quality, we decompose the CodeBLEU score into its four subcomponents in Figure 2 (e)–(h). Both AI-generated and AI-perturbed code show relatively low BLEU (Papineni et al., 2002) and weighted BLEU (Ren et al., 2020) scores, indicating limited n-gram overlap with their human counterparts. While the syntactic AST match and semantic data-flow (Guo et al., 2021) match scores of AI code exceed 0.5 on average, suggesting that despite token-level divergence, both AI-perturbed and AI-paraphrased code maintains a fair level of syntactic and semantic consistency with human code. More detailed statistics are presented in Appendix H.

## 4 Progressive Robustness Evaluation Scenarios

In this section, we present how *CodeMirage* systematically evaluates the robustness of state-of-the-art AI-generated code detectors across six progressively challenging tasks. Our goal is to clarify their practical reliability in real-world scenarios by identifying the conditions under which they remain effective, exposing their limitations, and offering guidance for informed selection and improvement. To this end, we design four evaluation settings: (1) in-distribution testing, (2) out-of-distribution (OOD) testing, (3) adversarial perturbations, and (4) hybrid testing that combines OOD shifts with adversarial perturbations. This progression reflects increasing levels of difficulty, enabling a systematic assessment that delineates applicability boundaries .

**In-Distribution Testing.** This setting evaluates each detector's stability when the training and test distributions align across **multi-LLMs** and **multi-programming-languages**. For each language, we pair the human-written training set with generated samples from a *single* LLM, and either train the detector on the combined data or select an optimal decision threshold. Evaluation is then conducted on the corresponding human-written test set, paired with test samples from the *same* LLM.

**Out-of-distribution (OOD) Testing.** This setting evaluates robustness to unseen generators (**cross-model**) and unseen languages (**cross-language**). In the cross-model case, for each language, we train and threshold the detector on human-written samples paired with AI-generated samples from a *single* LLM, and then test on the same human-written set paired with samples from all *other* LLMs. In the cross-language case, we train on one language and evaluate on human-written and AI-generated samples from the *same* LLM across all other languages. Results are averaged over unseen-model or unseen-language test sets separately to yield the final OOD performance.

**Adversarial Perturbation.** This setting evaluates robustness against adversarial perturbations while keeping the generating LLM fixed. We train and threshold the detector following the in-distribution configuration, but test on *perturbed* AI-generated code derived from the same LLM's original test samples. Our primary perturbation technique is paraphrasing, supplemented by three advanced attacks on a controlled subset of data to assess robustness against stronger adversaries.

**Hybrid Testing (OOD + Adversarial Perturbation).** This setting captures real-world scenarios where code is generated by unseen LLMs and subsequently perturbed. We follow the cross-model and cross-language testing protocols but pair human-written test samples with *perturbed* AI-generated samples from the other LLMs or languages. Detector performance is then averaged across these perturbed, unseen-model / unseen-language test sets to yield the hybrid result.

## 5 Benchmark Results and Insights

We conduct an extensive evaluation using *CodeMirage* in various scenarios and summarize the observations into eight findings. We present representative processed results in the main text and include the full experimental results in Appendix K.

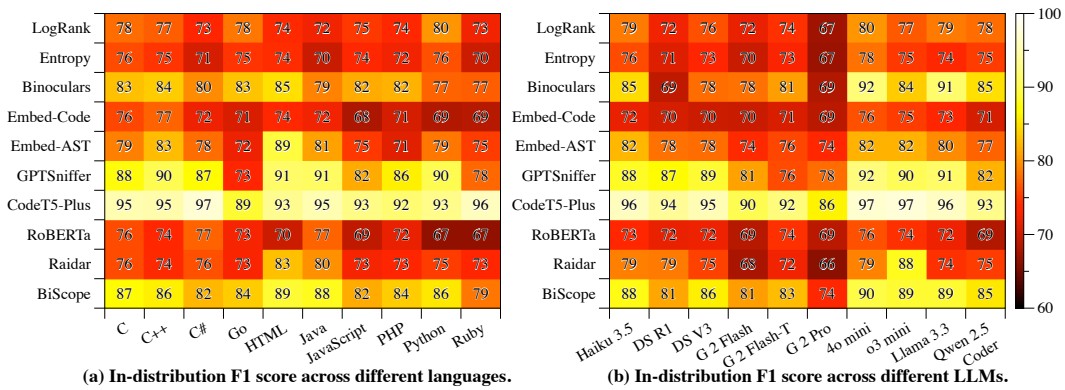

(a) **In-distribution F1 score across different languages.**    (b) **In-distribution F1 score across different LLMs.**

Figure 3: In-distribution performance (F1 score) comparison between all the baselines.

## 5.1 EVALUATION SETUP

**Detectors.** We evaluate ten state-of-the-art detectors spanning four categories. **Zero-shot detectors**: *LogRank* (Gehrmann et al., 2019), *Entropy* (Gehrmann et al., 2019; Lavergne et al., 2008), and *Binoculars* (Hans et al., 2024), which rely on token-rank or entropy-related features without training. **Embedding-based detectors**: following existing studies (Suh et al., 2025), we extract representations with the *CodeXEmbed-2B* model (Liu et al., 2024b) from either raw source code or its abstract-syntax tree (AST) and train a lightweight random forest (Breiman, 2001) classifier. **Fine-tuned detectors**: we include *GPTSniffer* (Nguyen et al., 2024; 2023), a variant built on the latest *CodeT5+* backbone (Wang et al., 2023), and a *RoBERTa* detector (Liu et al., 2019), with each fine-tuned on our training corpus. **Pretrained-LLM with downstream detector**: *Raidar* (Mao et al., 2024) and *BiScope* (Guo et al., 2024), extracting features via rewriting (Mao et al., 2024) and bi-directional cross entropy (Guo et al., 2024). More details are presented in Appendix I.

**Metrics.** To thoroughly assess the performance of the baseline detectors in different scenarios, we employ three evaluation metrics in our experiments, including the *F1 score*, *TPR@FPR=10%*, and *TPR@FPR=1%*. The *F1 score* balances precision and recall, providing an overall measure of detection accuracy without favoring AI-generated or human-written code samples. For each detector, we first identify the optimal decision threshold and then report its corresponding *F1 score*. The metric *TPR@FPR=10%* reports the true positive rate (TPR) when the false positive rate (FPR) is limited to 10%, representing scenarios that can tolerate a moderate number of false alarms. Conversely, *TPR@FPR=1%* measures the TPR at an FPR of only 1%, which is essential for applications where even a small fraction of false positives is unacceptable.

## 5.2 ASSESS THE ROBUSTNESS OF DETECTORS UNDER A WIDE SPECTRUM OF SCENARIOS

**In-distribution results are strong but potentially over-optimistic.** The in-distribution results are shown in Figure 3, with sub-figure (a) illustrating the results across different programming languages and (b) across different LLMs. Each value denotes the F-1 score averaged for all LLMs and langauges respectively. Observe that the overall detection performance is strong, but may be over-optimistic due to the overfitting problem.

> **Finding 1:** *Fine-tuning-based methods outperforms other types consistently under in-distribution setting.*

Fine-tuned detectors, e.g., GPTSniffer and CodeT5+, lead the pack, as the test data share the same distribution with the training data, potentially introduce overfitting problem. Zero-shot approaches, on the other hand, e.g., LogRank and Entropy, perform poorest, which makes sense given their limited feature extraction when confronted with the complexity of code.

> **Finding 2:** *Detection is Consistent across Programming Languages, with Common Languages Performing Slightly Better.*

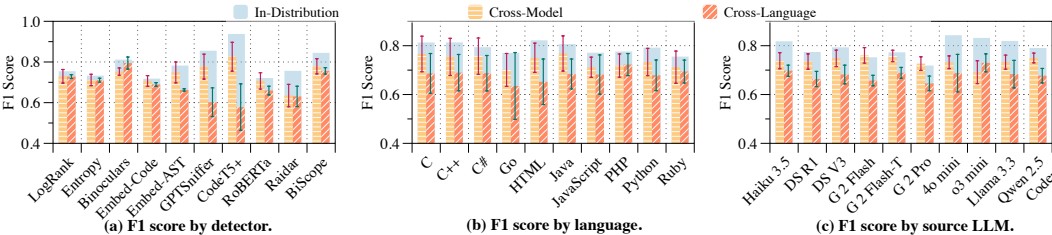

Figure 4: Comparison under cross-model and cross-language OOD configurations by detector, language, and source LLM.

We observe only slight performance differences across languages. Notably, less common languages lead to marginally lower performance. For example, detectors typically achieve higher F1 scores on C++ than Go or Ruby. This discrepancy arises because several detection methods, e.g., Biscope (Guo et al., 2024) and Raidar (Mao et al., 2024), rely on pre-trained LLMs for feature extraction. These models are pre-trained on large online corpora containing more examples of common languages (e.g., C++) than atypical ones (e.g., Ruby), resulting in stronger representations for the former. Hence detection performances are better detection on those common languages.

**Finding 3:** *Detection performance is largely consistent across LLMs, with smaller or less capable models being easier to detect.*

Among all generators, code produced by GPT-4o-mini is the easiest to detect, as reflected by consistently higher scores across all detectors. This suggests that its code style is more regular and distinctive, facilitating detection. Claude 3.5 Haiku and Llama 3.3 70B also yield relatively easy-to-detect outputs. In contrast, code generated by Gemini-2.0-Pro is the most difficult to detect. Overall, these results indicate that detection difficulty tends to increase with model scale and output diversity.

**OOD testing significantly reduce the performance.** Out-of-distribution results are shown in Figure 4, with sub-figure (a) presents the F1 scores across detectors and (b) (c) across languages and LLMs. We observe a constant significant degradation.

**Finding 4:** *Fine-tuned approaches exhibit the largest degradation under OOD testing, particularly in the cross-language setting, whereas zero-shot detectors show the opposite trend with stronger robustness.*

Fine-tuned detectors (e.g., GPTSniffer and CodeT5-Plus) achieve the strongest in-distribution F1 scores but suffer the steepest declines out of distribution, reflecting overfitting of high-capacity classifiers to the training corpus. In contrast, zero-shot methods remain markedly more stable under OOD shifts, while embedding-based and pretrained-LLM–with–downstream-detector approaches experience moderate degradation. ***This observation underscores the value of zero-shot detectors.*** Although their in-distribution performance is weaker, their robustness under OOD conditions makes them an important and promising research direction.

**Finding 5:** *Distribution shift induces a uniform, moderate performance drop across both languages and source LLMs, with no pronounced outliers.*

The error bars largely overlap and per-language/per-model differences are small, indicating that the loss is global rather than tied to specific syntax or generator families. Cross-language transfer is typically harder than cross-model.

**Adversarial perturbations consistently reduce detection performance.** Figure 5 reports F1 under clean (blue) and paraphrased (hatched teal) inputs by detector (a), language (b), and source LLM (c). We observe a near-uniform degradation across all settings, with overlapping error bars and no systematic outliers.

**Finding 6:** *Semantics-preserving paraphrases induce a broad, uniform performance drop across languages and source LLMs.*

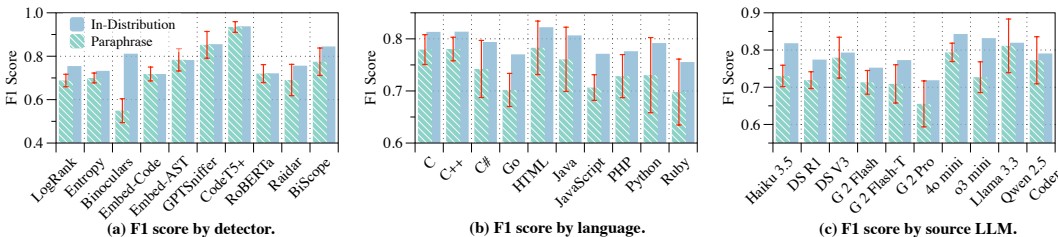

Figure 5: Comparison under adversarial perturb by detector, language, and source LLM.

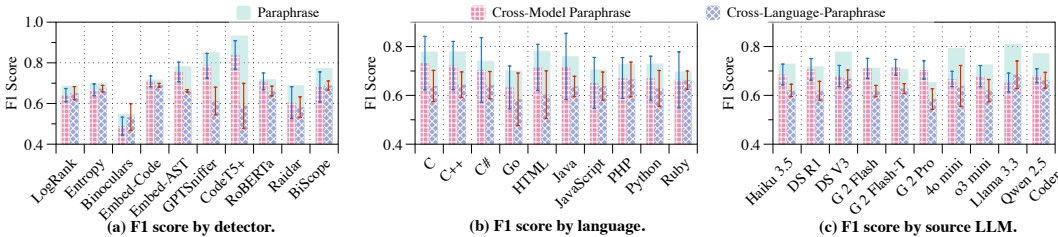

Figure 6: Comparison under cross-model perturb and cross-language perturb adversarial configurations by detector, language, and source LLM.

> **Finding 7:** *Perturbation conducted by reasoning models usually lead to a larger performance drop.*

We observe that for non-reasoning models (DeepSeek V3, GPT4o mini, Llama 3.3 70B, and Qwen 2.5 Coder 32B), paraphrasing has minimal impact on performance. In contrast, reasoning models (e.g., GPT o3 mini) lead to a more pronounced decline of detector's performance. This likely stems from their stronger comprehension abilities: they better interpret paraphrased inputs and adjust outputs to match human-style reasoning, making any deviations more evident after paraphrasing.

**Hybrid setting is the most challenging.** Figure 6 evaluates detectors under *both* paraphrase perturbation and OOD transfer — our closest proxy to real deployment where both the source generator and style are unknown. The baseline bars correspond to the paraphrase-only scores; adding OOD shift produces an additional, consistent drop across detectors, languages, and LLMs.

> **Finding 8:** *The hybrid setting compounds distribution shifts and adversarial perturbations, yielding uniform degradation across languages and source LLMs (with no clear outliers), and fine-tuned detectors suffer the largest additional loss.*

The pronounced degradation under OOD perturbation underscores the fragility of existing AI-generated code detectors in real-world scenarios. Our TPR-at-low-FPR results reveal a similar trend, as shown in Appendix J. More obvious evidence is also observed under three advanced attacks on a controlled subset of data, reported in Appendix G.

To further validate the robustness of our conclusions, we conduct comprehensive $t$-tests across all tasks and settings. The results, presented in Appendix L, consistently support our key findings. We also discuss limitations and future work in Appendix M.

## 6 CONCLUSION

In this paper, we present *CodeMirage*, a large-scale benchmark for AI-generated code detection, spanning 10 widely used programming languages and approximately 210,000 samples. The dataset includes human-written code as well as AI-generated and perturbed variants produced by 10 state-of-the-art production-level LLMs, with rigorous quality control to ensure reliability. We evaluate 10 representative detectors across four methodological categories under six progressively challenging tasks and four evaluation configurations, conducting extensive analyses that reveal the strengths and limitations of each approach. The breadth and depth of *CodeMirage* provide a solid foundation for advancing the development of more robust and generalizable code detectors.

## ETHICAL STATEMENT

As a benchmark paper, we acknowledge the risk of inadvertently collecting private or low-quality code from the web. To mitigate this, we do not directly source human-written code from GitHub. Instead, we rely on the well-established CodeParrot GitHub-Code-Clean CodeParrot (2022b) dataset. Additional ethical considerations, bias statements, and details about the human source code are provided with figures in Appendix A.

Although **CodeMirage** encompasses a diverse range of programming languages, LLMs, and adversarial scenarios, it is primarily designed for benchmarking detection performance in general-purpose coding contexts. Applying it to specialized domains (e.g., student programming assignments) may require domain adaptation or additional data collection to ensure proper alignment with the target setting. ***Importantly, we do not recommend using CodeMirage for directly training and deploying detectors in real-world applications.*** The primary goal of **CodeMirage** is to enable rigorous and realistic evaluation, allowing researchers to better identify detector limitations and enhance robustness prior to deployment.

We also provided a statement regarding the use of large language models (LLMs) in Appendix N.

## REPRODUCIBILITY STATEMENT

To ensure reproducibility, we provide detailed step-by-step descriptions of our data construction pipeline and evaluation methodology within the paper. We will release all benchmark data, and evaluation code under an open-source license upon acceptance. In addition, we include implementation details, hyperparameter configurations, and dataset statistics in the appendices to facilitate faithful replication of our results. Together, these resources are intended to enable the community to fully reproduce our experiments and extend **CodeMirage** to future studies.

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

APPENDIX

To further support and validate our *CodeMirage* benchmark, we provide the following supplementary materials:

- Appendix A: Detailed ethical and bias statement for human code collection and preprocessing in *CodeMirage*.
- Appendix B: Detailed descriptions of the production-level LLMs used in *CodeMirage* and their corresponding generation settings.
- Appendix C: Prompts used in the code summarization phase and representative examples.
- Appendix D: The rationale behind our BLEU-based data filtration.
- Appendix E: Prompts used in the code generation phase and representative examples.
- Appendix F: Domain-specific transformation rules, prompts, used in the code paraphrasing phase with representative examples.
- Appendix G: Introduction and results with three advanced perturbation techniques.
- Appendix H: Comprehensive statistics and distributions of the *CodeMirage* dataset.
- Appendix I: Detailed descriptions of the baseline detectors included in our evaluation.
- Appendix J: Supplementary results based on *TPR@FPR* metrics.
- Appendix K: Extended and detailed experimental results across all evaluation settings.
- Appendix L: T-test across all the tasks' results to show the steadiness of our key findings.
- Appendix M: Additional discussion on the limitations and future improvement directions.
- Appendix N: LLM usage declaration.

## A    ADDITIONAL ETHICAL INFORMATION FOR HUMAN CODE IN *CodeMirage*

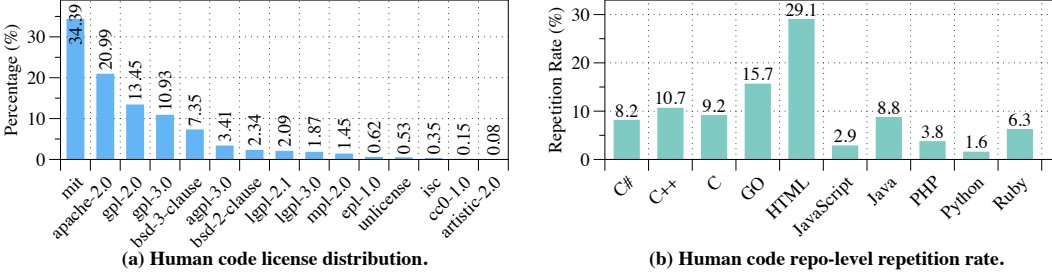

(a) **Human code license distribution.**    (b) **Human code repo-level repetition rate.**

Figure 7: *CodeMirage*'s human source code license and diversity information.

As described in our main text and ethical statement, *CodeMirage* does not directly collect human-written code from GitHub. Instead, it relies on the CodeParrot GitHub-Code-Clean dataset (CodeParrot, 2022b), a curated and moderated subset from a larger public GitHub dataset maintained by Google and GitHub. This ensures a high-quality and privacy-conscious data source.

All human-written code samples in our *CodeMirage* are hence from open-source projects with licenses permitting redistribution and research use. As shown in Figure 7 (a), the license distribution includes widely used licenses such as MIT, Apache-2.0, and GPL variants. For licenses requiring license notices (e.g., GPL-3.0), we retain the original license headers.

Besides, the ten programming languages in *CodeMirage* were selected based on their frequency in the GitHub-Code-Clean dataset (CodeParrot, 2022b), which contains over 850GB of curated code from open-source projects on GitHub. We chose the top 10 most widely used languages in this corpus, prioritizing those with high practical relevance and widespread exposure to LLM-generated code in current development workflows.

Table 2: Detailed configurations of the production-level LLMs used in *CodeMirage*.

| LLM Name | API / Model Path | Hyper-Parameter |
|---|---|---|
| Claude-3.5-Haiku (Anthropic, 2024) | `Anthropic/claude-3-5-haiku-20241022` | temperature = 1.0 |
| GPT-4o-mini (OpenAI, 2024) | `OpenAI/gpt-4o-mini-2024-07-18` | temperature = 1.0 |
| GPT-o3-mini (OpenAI, 2025) | `OpenAI/o3-mini-2025-01-31` | temperature = 1.0 reasoning_effort = medium |
| Gemini-2.0-Flash (Pichai et al., 2024) | `Google/gemini-2.0-flash` | temperature = 1.0 |
| Gemini-2.0-Flash-Thinking (Kane, 2025) | `Google/gemini-2.0-flash-thinking-exp-01-21` | temperature = 1.0 |
| Gemini-2.0-Pro (Kavukcuoglu, 2025) | `Google/gemini-2.0-pro-exp-02-05` | temperature = 1.0 |
| DeepSeek-V3 (Liu et al., 2024a) | `deepseek-ai/DeepSeek-V3` | temperature = 1.0 |
| DeepSeek-R1 (Guo et al., 2025) | `deepseek-ai/DeepSeek-R1` | temperature = 1.0 |
| Llama-3.3-70B (Meta, 2024) | `meta-llama/Llama-3.3-70B-Instruct` | temperature = 0.6 |
| Qwen-2.5-Coder-32B (Hui et al., 2024) | `Qwen/Qwen2.5-Coder-32B-Instruct` | temperature = 0.7 |

We acknowledge that CodeMirage may not yet cover certain emerging or underrepresented languages, as expanding to all possible languages would require substantial additional resources. However, to the best of our knowledge, our language coverage already exceeds that of existing benchmarks in this area, as shown in Table 1.

Moreover, we mitigate repository-specific bias by sampling from a broad set of projects. As shown in Figure 7 (b), our repo-level repetition rate analysis reveals that, for most languages, fewer than 10% of human-written code samples originate from the same repository. In total, the 10,000 human-written code files in *CodeMirage* are sourced from 9,037 unique repositories, effectively minimizing the risk of dominance by any single repository.

## B    DETAILS OF GENERATIVE MODELS AND GENERATION SETTINGS

In *CodeMirage*, we leverage ten production-level LLMs from six leading companies to generate code samples, covering the majority of LLMs commonly used for real-world coding tasks. Among these ten models, four are open-source and three are designed with reasoning capabilities. Specifically, *CodeMirage* includes GPT-4o-mini (OpenAI, 2024), o3-mini (OpenAI, 2025), Claude-3.5-Haiku (Anthropic, 2024), Gemini-2.0-Flash (Pichai et al., 2024), Gemini-2.0-Flash-Thinking-Experimental (Kane, 2025), Gemini-2.0-Pro-Experimental (Kavukcuoglu, 2025), DeepSeek-V3 (Liu et al., 2024a), DeepSeek-R1 (Guo et al., 2025), Llama-3.3-70B (Meta, 2024), and Qwen-2.5-Coder-32B (Hui et al., 2024). We access all ten LLMs via API-based services with default temperatures. Detailed configurations and generation settings for these models are presented in Table 2. For key generation hyper-parameters such as `temperature` and `reasoning_effort`, we use either default values or officially recommended settings to reflect realistic usage. Importantly, we avoid setting `temperature` to zero, as doing so would produce overly deterministic outputs that are easier to detect. Instead, we adopt general-purpose settings for high-quality while more diverse and less predictable code generation.

## C    ADDITIONAL DETAILS OF AI CODE SUMMARIZATION

To generate high-quality and representative summaries that comprehensively describe the characteristics of a code sample while preventing the leakage of concrete code, we design a structured summarization prompt covering eight key aspects. We then prompt the LLMs to act as summarizers, generating summaries based on the input code file using this carefully crafted prompt. The full summarization prompt used in *CodeMirage* is as follows:

> **💡 Summarization Prompt**
>
> Analyze the provided code snippet and generate a concise and informative description of its functionality, purpose, and design. Avoid directly including or mirroring the given code. Focus on abstracting the logic, functionality, and intent.
>
> Follow the output format:
>
> 1. **Purpose:** A high-level summary of what the code is intended to achieve.

2. **Functionality:** Describe the main tasks performed by the code, including inputs, outputs, and their roles, without referencing exact code or variable names.

3. **Logic Overview:** Explain the key logic, algorithms, or patterns conceptually, avoiding specific code structures or syntax.

4. **Key Features:** Highlight unique approaches or techniques without mentioning explicit implementation details.

5. **Key Libraries:** Include the key libraries used in the code snippet.

6. **Key Functions:** Include the key function names used in the code snippet.

7. **Key Class, Structure, Variable:** Include the key class, structure, and variable names used in the code snippet.

8. **Contextual Notes (optional):** Any assumptions, dependencies, or inferred background information.

Below, we present a concrete Python code example with the summary generated by GPT-o3-mini (OpenAI, 2025), illustrating the consistency and descriptive quality of the summarization process in *CodeMirage*.

---

**⟨⧸⟩ Python Summary Example**

**Human-Written Code**

```python
import logging
import voluptuous as vol
from homeassistant.components.binary_sensor import
    (BinarySensorDevice, PLATFORM_SCHEMA, SENSOR_CLASSES_SCHEMA)
from homeassistant.components import enocean
from homeassistant.const import (CONF_NAME, CONF_ID, CONF_SENSOR_CLASS)
import homeassistant.helpers.config_validation as cv

_LOGGER = logging.getLogger(__name__)
DEPENDENCIES = ['enocean']
DEFAULT_NAME = 'EnOcean binary sensor'

PLATFORM_SCHEMA = PLATFORM_SCHEMA.extend({
    ...
})

def setup_platform(hass, config, add_devices, discovery_info=None):
    ...
class EnOceanBinarySensor(enocean.EnOceanDevice, BinarySensorDevice):
    def __init__(self, dev_id, devname, sensor_class):
        ...
    @property
    def name(self):
        ...
    @property
    def sensor_class(self):
        ...
    def value_changed(self, value, value2):
        ...
...
```

- - - - - - - - - - - - - - - - - - - - - - - - - - - - - - - - - - - - - - - - -

**GPT-o3-mini Generated Summary**

1. **Purpose:** The code integrates binary sensors that communicate using an EnOcean protocol into a larger home automation framework. It enables sensor events to be received, processed, and dispatched within the automation system.

2. **Functionality:** The implementation defines a platform component for binary sensors. It validates configuration inputs, creates ...

3. **Logic Overview:** The code establishes a configuration schema to ensure correct setup parameters. It then creates a ...

4. **Key Features:** Validates configuration using a dedicated schema to ensure data consistency. Leverages inheritance to ...

5. **Key Libraries:** Logging, Voluptuous for configuration validation, ...

6. **Key Functions:** ... An event handler method that processes changes in sensor values and triggers events ...

7. **Key Class, Structure, Variable:** A sensor class that ... , Variables for device identification, sensor naming, ...

8. **Contextual Notes:** The design assumes integration within a home automation ecosystem where components communicate via ...

For more examples and summaries across different programming languages and generative models, please refer to our *CodeMirage* dataset.

# D ADDITIONAL DETAILS OF THE BLEU-BASED DATA FILTRATION

To filter potentially memorized AI-generated code, we use a BLEU-based inspector and set a conservative threshold of 0.5. This section explains the rationale behind this choice.

Specifically, to filter potentially memorized AI-generated code, we use a BLEU-based inspector and set a conservative threshold of 0.5. This section explains the rationale behind this choice.

**⏣ Human-Written Code Example**

```python
from typing import List

def has_close_elements(numbers: List[float], threshold: float) -> bool:
    """ Check if in given list of numbers, are any two numbers closer to each other than
    given threshold.
    >>> has_close_elements([1.0, 2.0, 3.0], 0.5)
    False
    >>> has_close_elements([1.0, 2.8, 3.0, 4.0, 5.0, 2.0], 0.3)
    True
    """
    for idx, elem in enumerate(numbers):
        for idx2, elem2 in enumerate(numbers):
            if idx != idx2:
                distance = abs(elem - elem2)
                if distance < threshold:
                    return True

    return False
```

**⏣ AI-generated Code Example**

**BLEU = 0.57 (Filtered Out)**

```python
from typing import List

def has_close_elements(numbers: List[float], threshold: float) -> bool:
    """
    Check if any two elements in the list are closer than the specified threshold.

    Args:
        numbers: List of floating-point numbers to compare
        threshold: Maximum allowed distance between two numbers

    Returns:
        Boolean indicating presence of close elements

    >>> has_close_elements([1.0, 2.0, 3.0], 0.5)
    False
    >>> has_close_elements([1.0, 2.1, 3.0], 1.0)
    True
    """
    for idx, elem in enumerate(numbers):
        for idx2, elem2 in enumerate(numbers):
            if idx != idx2:
                distance = abs(elem - elem2)
                if distance < threshold:
                    return True
    return False
```

As shown above, this AI-generated example shares substantial overlap in structure, variable naming, and logic with the human-written version. Despite being technically valid, it lacks sufficient novelty and is filtered out to avoid near-duplicates.

**AI-generated Code Example**

**BLEU = 0.45 (Kept)**

```python
from typing import List

def has_close_elements(numbers: List[float], threshold: float) -> bool:
    """
    Determines if any two elements in the list are closer than the threshold.

    Args:
        numbers: List of floating point numbers to check
        threshold: Maximum allowed distance between elements

    Returns:
        Boolean indicating presence of closely positioned numbers

    >>> has_close_elements([1.0, 2.0, 3.0], 1.5)
    False
    >>> has_close_elements([1.0, 2.1, 3.0], 1.0)
    True
    """
    for idx, elem in enumerate(numbers):
        for idx2, elem2 in enumerate(numbers[idx+1:], start=idx+1):
            distance = abs(elem - elem2)
            if distance < threshold:
                return True
    return False
```

In contrast, this version introduces meaningful variation, such as an optimized iteration and rewritten documentation. While functionally correct, it is not semantically identical to the original, and thus retained.

Furthermore, as shown in Figure 2 (e)-(f), the majority of AI-generated and paraphrased samples naturally fall well below the 0.5 BLEU threshold. This indicates that our generation pipeline already reduces memorization risk, and the BLEU filter acts as a conservative safeguard rather than a strict enforcement mechanism.

# E    ADDITIONAL DETAILS OF AI CODE GENERATION

After generating a comprehensive yet concise summary of the human-written code, we prompt LLMs to produce corresponding AI-generated versions. To preserve structural consistency, we explicitly include the statistical properties (*e.g.*, line of code and character length) of the original code in the prompt. The complete code generation prompt used in *CodeMirage* is presented below:

**Code Generation Prompt**

Generate a {Language} code snippet with around {Line_of_Code} lines and {Char_Length} character length following the description. Only output the code, no any other information.

{Summary}

The generated code is then passed through a rule-based inspector, as described in subsection 3.2, to filter out overly short, overly long, or null outputs. Below, we show a Python code example generated by GPT-o3-mini (OpenAI, 2025), based on the summary presented in Appendix C:

**AI-Generated Python Example**

**GPT-o3-mini Code Generation**

```python
import logging
import voluptuous as vol
from homeassistant.const import CONF_NAME, CONF_DEVICE_ID
...
DOMAIN = "enocean_binary_sensor"
```

```
1134
1135     CONF_SENSOR_TYPE = "sensor_type"
1136     ...
         class EventBus:
1137         def fire(self, event_type, event_data):
1138             ...
         class EnOceanDevice:
1139         def __init__(self, device_id):
1140             ...
             def send_message(self, message):
1141             ...
1142     ...
         class BinarySensorDevice:
1143         ...
1144     class EnOceanBinarySensor(EnOceanDevice, BinarySensorDevice):
             ...
1145     ...
1146     if __name__ == "__main__":
             logging.basicConfig(level=logging.DEBUG)
1147         hass = Hass()
1148         config = {
                 CONF_DEVICE_ID: "enocean_001",
1149             ...
1150         }
             setup_platform(hass, config)
1151
1152
```

We observe that though the detailed implementation of the AI-generated code differs from the human-written version, it employs the same key libraries and similar variables to achieve comparable functionality, illustrating the quality of the AI-generated code. More examples are presented in ***CodeMirage*** dataset.

## F  ADDITIONAL DETAILS OF AI CODE PARAPHRASING

We further evaluate the robustness of detectors under adversarial paraphrasing, a prevalent and practical challenge in real-world applications. Our adversarial setup is grounded in prompt-based paraphrasing, which represents one of the most accessible and commonly used evasion strategies in practice. Unlike prior works on natural language paraphrasing (Krishna et al., 2023; Sadasivan et al., 2023), which rely on generic and concise prompts, we propose a code-specific paraphrasing prompt tailored to the programming domain. This prompt incorporates seven transformation strategies specifically designed for source code. The full prompt is provided below:

> **💡 Paraphrasing Prompt**
>
> Transform the given code to bypass AI-generated code detectors by mimicking human coding styles. Make structural, stylistic, and naming changes while preserving exact functionality. Apply the following:
>
> 1. Rename variables, functions, and classes with meaningful, human-like names.
>
> 2. Adjust formatting (indentation, spacing, line breaks) and reorganize imports logically.
>
> 3. Rewrite logic with equivalent constructs (*e.g.*, refactor loops, conditionals).
>
> 4. Replace common standard library functions with alternatives or custom helpers.
>
> 5. Vary expressions (*e.g.*, 'a + b' → 'sum([a, b])') and simplify inline logic into reusable functions.
>
> 6. Transform literals (*e.g.*, single vs. double quotes, '10' → '0xA') and reorder independent code blocks.
>
> 7. Add subtle redundancies (*e.g.*, intermediate variables) to resemble human-written code.
>
> Keep the line of code and character length similar to the original code and only output the transformed code. No any other additional text and information.

We then apply the paraphrasing prompt to the AI-generated code sample and obtain its paraphrased version. Below, we present the paraphrased code example produced by GPT-o3-mini (OpenAI, 2025), based on the AI-generated example provided in Appendix E.

**AI-Paraphrased Python Example**

**GPT-o3-mini Paraphrased Code**

```python
import logging
import voluptuous as schema_validator

from homeassistant.const import CONF_NAME, CONF_DEVICE_ID
import homeassistant.helpers.config_validation as config_val

logger = logging.getLogger(__name__)
SENSOR_DOMAIN = 'enocean_binary_sensor'
KEY_SENSOR_KIND = 'sensor_type'
...
SensorConfigSchema = schema_validator.Schema({
    ...
})

class SimulatedEventBus:
    def trigger(self, event_category, particulars):
        ...
    ...

class BaseEnOceanDevice:
    def __init__(self, dev_identifier):
        ...
    def dispatch_message(self, msg_payload):
        ...
class BasicBinarySensor:
    ...
class EnOceanSensorClient(BaseEnOceanDevice, BasicBinarySensor):
    ...
...
if __name__ == '__main__':
    logging.basicConfig(level=logging.DEBUG)
    simulated_hass = SimulatedHomeAssistant()
    input_configuration = {
        ...
    }
    initialize_platform(simulated_hass, input_configuration)
```

Compared to the original AI-generated code, the paraphrased version uses different aliases for imported libraries, introduces redundant classes and variables, and modifies function and class names with different implementations, while preserving the overall program functionality. Additional examples can be found in the full *CodeMirage* dataset.

## G ROBUSTNESS EXPLORATION WITH MORE ADVANCED ATTACKS

Additional to the paraphrasing technique introduced in the main text, we further incorporate three advanced adversarial strategies to stress test AI-generated code detectors on a controlled subset of data. These techniques emulate more sophisticated manipulations that could realistically be employed by motivated adversaries, thereby enhancing both the realism and diversity of our benchmark.

**Multi-Round Paraphrasing:** We design an LLM-based feedback loop in which an auxiliary model evaluates whether paraphrased code satisfies predefined transformation rules. When violations are detected, the auxiliary model provides refinement suggestions, and another paraphrasing round is triggered. This iterative process yields increasingly sophisticated and evasive adversarial samples.

**DeepWordBug (Gao et al., 2018):** We adapt this well-established obfuscation method, originally proposed for text classification, to the code detection setting. Specifically, we implement a multi-round variant that directly targets our RoBERTa-based detector, thereby simulating realistic, targeted attacks on learning-based detection models.

**AST-based Perturbation:** As an untargeted variant of DeepWordBug, this technique applies character-level perturbations to non-functional code elements (e.g., comments, function names) identified through AST analysis. Importantly, these transformations do not alter program semantics, but they increase surface-level variability and obfuscation.

Table 3: Performance of baseline detectors under three advanced attacks and their combinations.

| Attack | LogRank | Entropy | Binoculars | Embed- Code | Embed- AST | GPTSniffer | CodeT5+ | RoBERTa | Raidar | BiScope |
|---|---|---|---|---|---|---|---|---|---|---|
| One-round Paraphrasing | 0.5271 | 0.5858 | 0.1914 | 0.7112 | 0.8314 | 0.9398 | 0.9684 | 0.9495 | 0.8243 | 0.6514 |
| Multi-round Paraphrasing | 0.4516 | 0.5161 | 0.1253 | 0.7112 | 0.8314 | 0.9630 | 0.9769 | 0.9532 | 0.8464 | 0.5676 |
| DeepWordBug | 0.3713 | 0.4531 | 0.5104 | 0.7081 | 0.8071 | 0.7140 | 0.9474 | 0.6667 | 0.6874 | 0.4871 |
| Paraphrasing + DeepWordBug | 0.1907 | 0.2659 | 0.0329 | 0.7097 | 0.8159 | 0.6550 | 0.9456 | 0.6667 | 0.6033 | 0.2398 |
| AST-based Perturbation | 0.0364 | 0.0514 | 0.0916 | 0.7081 | 0.8228 | 0.9000 | 0.9562 | 0.7721 | 0.4279 | 0.4507 |
| Paraphrasing + AST-based Perturbation | 0.0074 | 0.0099 | 0.0144 | 0.7050 | 0.8331 | 0.8507 | 0.9384 | 0.6667 | 0.2021 | 0.5120 |

Table 4: *CodeMirage*'s data quantity statistics across different LLMs and programming languages.

| LLM | Paraphrase | Python | Java | JavaScript | C++ | C | C# | Go | Ruby | PHP | HTML |
|---|---|---|---|---|---|---|---|---|---|---|---|
| Human | ✗ | 1,000 | 1,000 | 1,000 | 1,000 | 1,000 | 1,000 | 1,000 | 1,000 | 1,000 | 1,000 |
| Claude-3.5-Haiku | ✗ | 1,000 | 1,000 | 1,000 | 1,000 | 1,000 | 1,000 | 1,000 | 1,000 | 1,000 | 1,000 |
| | ✓ | 1,000 | 1,000 | 1,000 | 1,000 | 1,000 | 1,000 | 1,000 | 1,000 | 1,000 | 1,000 |
| DeepSeek-R1 | ✗ | 1,000 | 1,000 | 1,000 | 1,000 | 1,000 | 1,000 | 1,000 | 1,000 | 1,000 | 1,000 |
| | ✓ | 1,000 | 1,000 | 1,000 | 999 | 1,000 | 999 | 1,000 | 1,000 | 1,000 | 1,000 |
| DeepSeek-V3 | ✗ | 1,000 | 1,000 | 1,000 | 1,000 | 1,000 | 1,000 | 1,000 | 1,000 | 1,000 | 1,000 |
| | ✓ | 1,000 | 1,000 | 1,000 | 1,000 | 1,000 | 1,000 | 1,000 | 1,000 | 1,000 | 1,000 |
| Gemini-2.0-Flash | ✗ | 1,000 | 1,000 | 1,000 | 1,000 | 1,000 | 1,000 | 1,000 | 1,000 | 1,000 | 1,000 |
| | ✓ | 1,000 | 1,000 | 1,000 | 1,000 | 1,000 | 1,000 | 1,000 | 1,000 | 1,000 | 1,000 |
| Gemini-2.0-Flash-Thinking | ✗ | 1,000 | 1,000 | 1,000 | 1,000 | 1,000 | 1,000 | 1,000 | 1,000 | 1,000 | 1,000 |
| | ✓ | 1,000 | 1,000 | 1,000 | 1,000 | 1,000 | 1,000 | 1,000 | 1,000 | 1,000 | 1,000 |
| Gemini-2.0-Pro | ✗ | 1,000 | 1,000 | 1,000 | 1,000 | 1,000 | 998 | 1,000 | 1,000 | 998 | 999 |
| | ✓ | 1,000 | 1,000 | 1,000 | 1,000 | 1,000 | 998 | 1,000 | 1,000 | 998 | 999 |
| GPT-4o-mini | ✗ | 1,000 | 1,000 | 1,000 | 1,000 | 1,000 | 1,000 | 1,000 | 1,000 | 1,000 | 1,000 |
| | ✓ | 1,000 | 1,000 | 1,000 | 1,000 | 1,000 | 1,000 | 1,000 | 1,000 | 1,000 | 1,000 |
| Llama-3.3-70B | ✗ | 1,000 | 1,000 | 1,000 | 1,000 | 1,000 | 1,000 | 1,000 | 1,000 | 1,000 | 1,000 |
| | ✓ | 1,000 | 1,000 | 1,000 | 1,000 | 1,000 | 1,000 | 1,000 | 1,000 | 1,000 | 1,000 |
| GPT-o3-mini | ✗ | 1,000 | 1,000 | 1,000 | 1,000 | 1,000 | 1,000 | 1,000 | 1,000 | 1,000 | 1,000 |
| | ✓ | 1,000 | 1,000 | 1,000 | 1,000 | 1,000 | 1,000 | 1,000 | 1,000 | 1,000 | 1,000 |
| Qwen-2.5-Coder-32B | ✗ | 1,000 | 1,000 | 1,000 | 1,000 | 1,000 | 1,000 | 1,000 | 1,000 | 1,000 | 1,000 |
| | ✓ | 1,000 | 1,000 | 1,000 | 1,000 | 1,000 | 1,000 | 1,000 | 1,000 | 1,000 | 1,000 |

Beyond these single techniques, we also construct hybrid adversarial attacks by combining one-round paraphrasing with either DeepWordBug or AST-based perturbations, thereby producing more complex and challenging examples. All experiments are conducted on Python code generated by `o3-mini`, with results presented in Table 3.

The results demonstrate that these advanced attacks substantially degrade detection performance across all baselines compared to one-round paraphrasing alone. Detectors that rely heavily on pre-trained language model signals — such as LogRank, Entropy, and BiScope — are particularly vulnerable, while embedding-based detectors exhibit greater resilience. Notably, CodeT5+ achieves the highest robustness overall.

Importantly, these findings reinforce our core observation: current detectors lack robustness against adversarial manipulations. Even lightweight paraphrasing already causes a marked performance drop, and the addition of stronger adversarial strategies further highlights their fragility under realistic and evolving attack scenarios.

# H ADDITIONAL STATISTICS OF *CodeMirage*'S DATASET

In subsection 3.4, we present detailed data quality statistics of the *CodeMirage* dataset across eight metrics. Here, we additionally report the dataset's quantity statistics, as summarized in Table 4. For both human-written code and most AI-perturbed code, we collect or synthesize 1,000 samples per programming language, with a standard 700/300 split for training and testing, respectively.

However, for some LLMs and specific languages, such as Gemini-2.0-Pro (Kavukcuoglu, 2025) on C#, we fall slightly short of the 1,000-sample target due to repeated generation refusals, typically caused by the model's built-in output filtering mechanisms. Despite these occasional gaps, the overall data quality and balance of the *CodeMirage* dataset remain unaffected.

Furthermore, we also evaluate the impact of three additional advanced adversarial attacks and their combinations on a representative subset of the dataset. Details are provided in Appendix G.

# I    ADDITIONAL DETAILS OF BASELINE DETECTORS

In this section, we provide additional introduction and implementation details for each of the ten baseline detectors evaluated in *CodeMirage*.

***LogRank* (Gehrmann et al., 2019) & *Entropy* (Lavergne et al., 2008).** These two baseline detectors represent classic zero-shot detection approaches that rely on pretrained LLMs. The underlying intuition is that LLMs are more familiar with AI-generated text or code, resulting in lower token-level log-rank or entropy values compared to human-written content. Both methods compute the average token-level statistic (log-rank or entropy) over the input, which is then used as the detection score. In *CodeMirage*, we implement these detectors using the state-of-the-art open-source pretrained model *Llama-3.2-3B-Instruct* (`meta-llama/Llama-3.2-3B-Instruct`) as the scoring backbone.

***Binoculars* (Hans et al., 2024).** *Binoculars* is a state-of-the-art zero-shot detector based on the insight that AI-generated text or code tends to receive more consistent scores across different LLMs than human-written content. To exploit this property, the method feeds the input simultaneously into two distinct LLMs and computes a novel *cross-perplexity* metric as the detection score. In *CodeMirage*, we adopt the official implementation[1] of *Binoculars* to ensure reproducibility and optimized performance.

***Embed-Code* (Suh et al., 2025) & *Embed-AST* (Suh et al., 2025).** These two embedding-based methods leverage pretrained code embedding models to extract semantic representations of entire code files. *Embed-Code* encodes the raw source code directly, while *Embed-AST* first parses the code into its abstract syntax tree (AST) using *tree-sitter*[2], and then encodes the AST. The embeddings are then passed to a supervised classifier for detection. In *CodeMirage*, we employ the latest *CodeXEmbed-2B* (Liu et al., 2024b) model as the embedding model and use a *Random Forest* (Breiman, 2001) classifier as the downstream detector.

***GPTSniffer* (Nguyen et al., 2023; 2024).** *GPTSniffer* is a state-of-the-art fine-tuning-based detector that leverages the code-related capability of *CodeBERT* (Feng et al., 2020). It is fine-tuned on a labeled dataset consisting of both human-written and AI-generated code samples, and evaluated on unseen test data. In *CodeMirage*, we adopt training hyperparameters consistent with prior work (Orel et al., 2025): 5 training epochs, a learning rate of 3e-4, weight decay of 1e-3, and a warmup ratio of 0.1. We train *GPTSniffer* on *CodeMirage*'s training set and evaluate on *CodeMirage*'s test set.

***CodeT5+* (Wang et al., 2023) & *RoBERTa* (Liu et al., 2019).** These two fine-tuning-based detectors follow the same training pipeline as *GPTSniffer*, but utilize different backbone models: the latest *CodeT5+* (Wang et al., 2023) and the classic *RoBERTa* (Liu et al., 2019). In *CodeMirage*, we use the same training hyperparameters and evaluation settings as those employed for *GPTSniffer* to ensure a fair comparison.

***Raidar* (Mao et al., 2024).** *Raidar* is based on the observation that LLMs tend to modify a greater proportion of human-written content compared to AI-generated content. It hence uses multiple prompts to instruct an LLM to rewrite the input and then computes a set of numerical features (*e.g.*, Bag-of-Words edit distance and Levenshtein score). These features are used to train a downstream classifier as the final detector. In *CodeMirage*, we adopt the latest *GPT-4.1-nano*[3] as the rewriting model, which is stronger than the original *GPT-3.5-Turbo* used in *Raidar*. We also follow the official implementation[4] to extract features and train the detection model.

***BiScope* (Guo et al., 2024).** *BiScope* is a state-of-the-art detector that leverages a pre-trained LLM to extract bi-directional entropy features, which are then used to train a lightweight downstream classifier. The bi-directional entropy is designed to capture both next-token prediction (forward entropy) and previous-token memorization (backward entropy) from the model's output logits. In *CodeMirage*, we use *Llama-3.2-3B-Instruct* (`meta-llama/Llama-3.2-3B-Instruct`) as the feature

---

[1]`https://github.com/ahans30/Binoculars`
[2]`https://github.com/tree-sitter/tree-sitter`
[3]`https://platform.openai.com/docs/models/gpt-4.1-nano`
[4]`https://github.com/cvlab-columbia/raidarllmdetect`

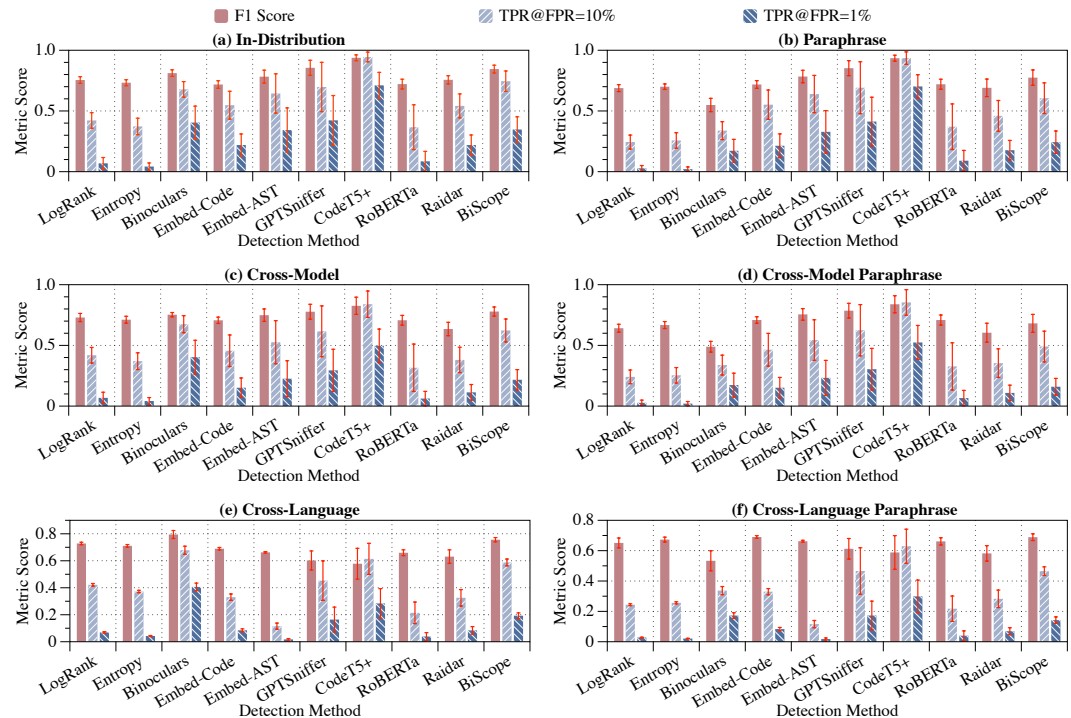

Figure 8: **Comparison Between Different Evaluation Metrics.** The bar charts illustrate the average F1 scores of baseline detectors on different LLMs across programming languages.

extractor for *BiScope*, consistent with the scoring model used in *LogRank* and *Entropy*. A *Random Forest* (Breiman, 2001) classifier is employed as the downstream detector.

## J    EVALUATION RESULTS OF ADDITIONAL METRICS

The results appear in Figure 8, where the x-axis lists the detection methods and the y-axis shows their metric values. As before, each bar reflects the mean performance across ten programming languages and ten LLMs, with error bars indicating one standard deviation. The figure is divided into six panels, each corresponding to a different evaluation task. Despite decent F1 scores across the board, all detectors suffer a dramatic drop in true-positive rate once the false-positive rate is constrained (e.g., TPR@FPR=1% is generally lower than 0.3), showing that they fail to catch enough positives under realistic, low false alarm requirements and are therefore less practical.

## K    ADDITIONAL EVALUATION RESULTS

In this section, we present the complete F1 scores of all baseline detectors evaluated across different LLMs and programming languages. Specifically, Figure 9 shows the results under the in-distribution task, while Figure 10 reports the scores under the adv. perturb task. Figure 11 illustrates the results under the cross-model task, and Figure 12 presents the scores under the cross-model perturb task. The results of cross-language task and cross-language perturb task are presented in Figure 13 and Figure 14, respectively.

These comprehensive results are consistent with the trends discussed in section 5, further validating the key findings derived from the ***CodeMirage*** evaluation.

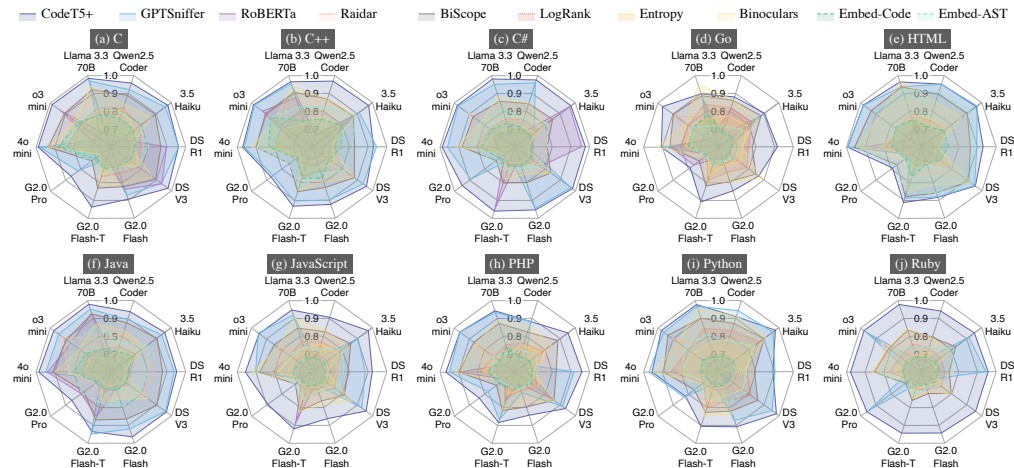

Figure 9: Complete F1 scores of all baseline detectors across various LLMs and programming languages under the in-distribution configuration.

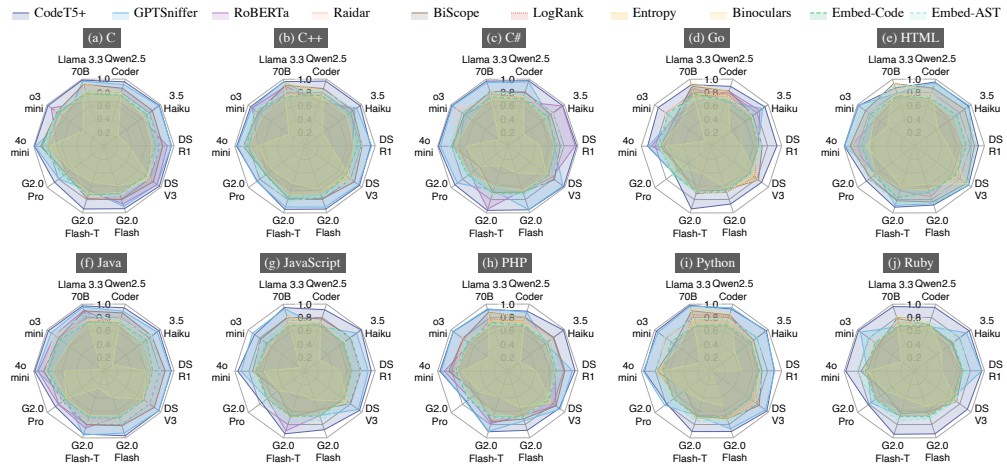

Figure 10: Complete F1 scores of all baseline detectors across various LLMs and programming languages under the paraphrase configuration.

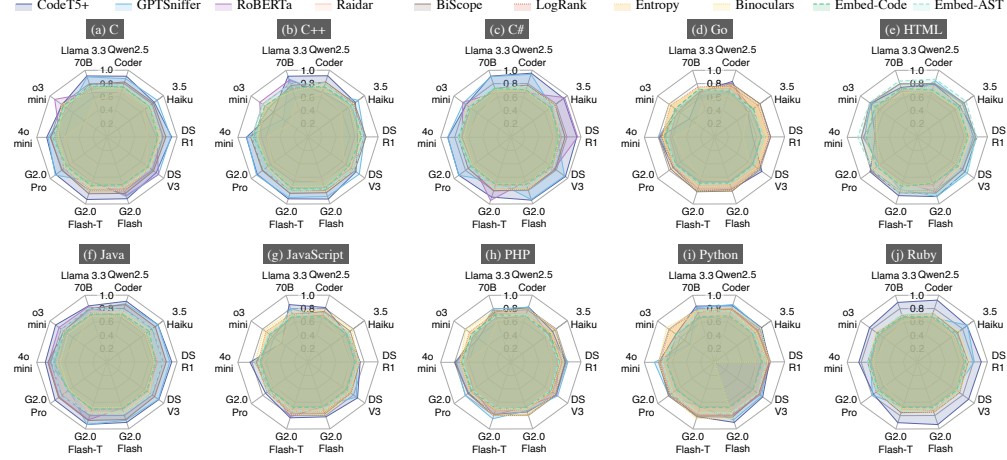

Figure 11: Complete F1 scores of all baseline detectors across various LLMs and programming languages under the cross-model configuration.

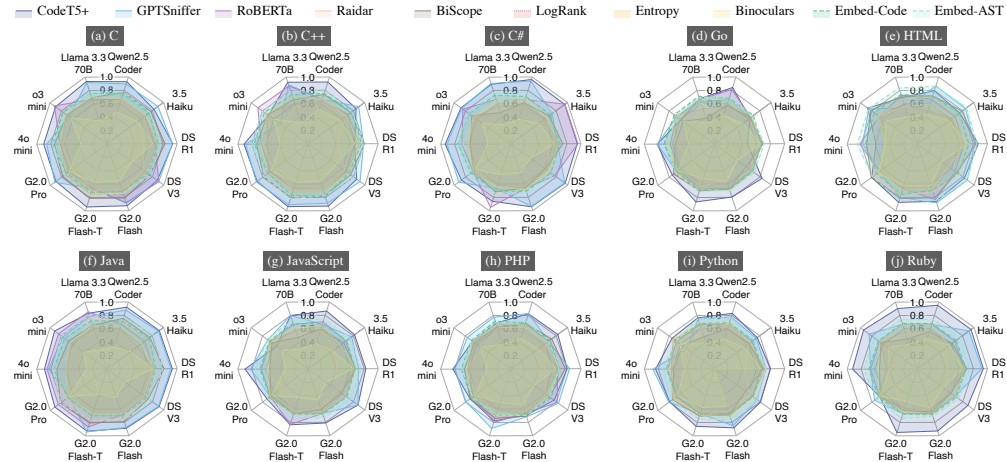

Figure 12: Complete F1 scores of all baseline detectors across various LLMs and programming languages under the cross-model paraphrase configuration.

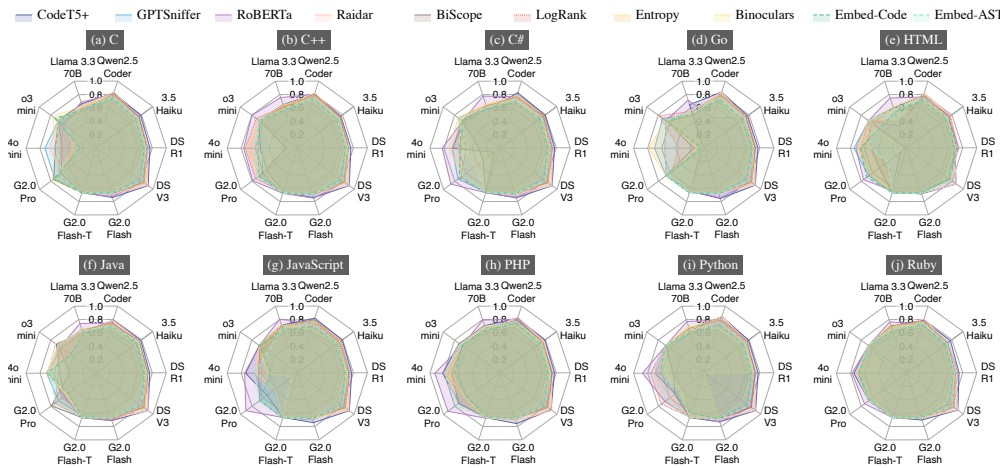

Figure 13: Complete F1 scores of all baseline detectors across various LLMs and programming languages under the cross-language configuration.

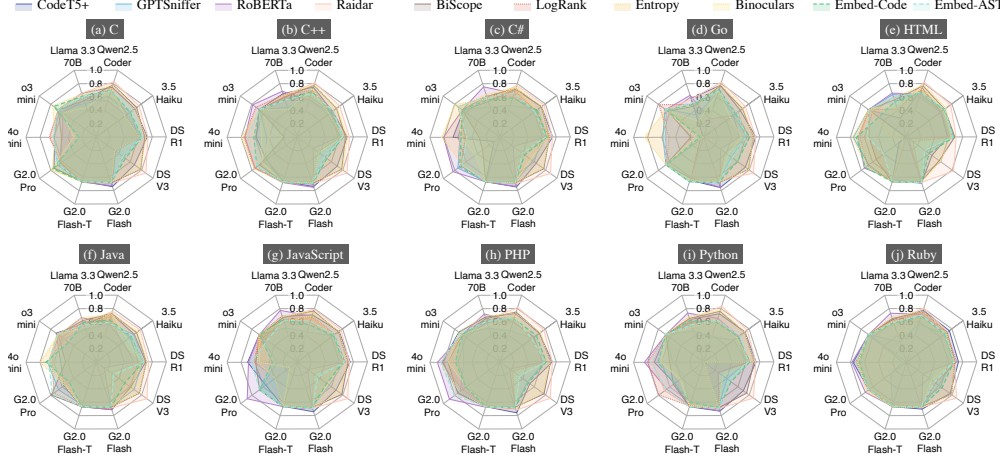

Figure 14: Complete F1 scores of all baseline detectors across various LLMs and programming languages under the cross-language paraphrase configuration.

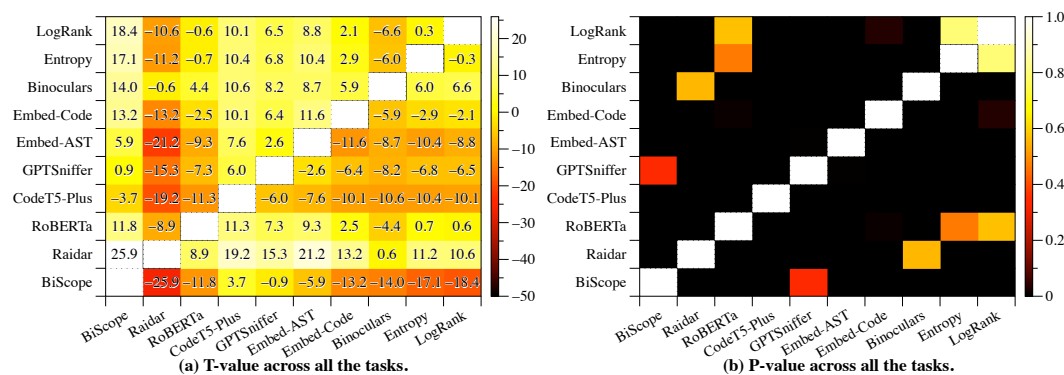

Figure 15: T-test result across all the evaluation tasks.

## L  T-TEST ACROSS ALL THE TASKS

To further evaluate how consistent and general of our key findings based on our evaluation, we further conduct the t-test across all the tasks' results, shown in Figure 15, across all 10 baselines over 600 distinct evaluation settings (10 languages × 10 generative models × 6 tasks). This analysis quantifies the statistical significance of observed performance differences. The results align well with our key findings in the main text. For example, fine-tuned methods such as CodeT5-Plus and GPTSniffer consistently outperform other baselines, as reflected in large-magnitude t-values and very low p-values when compared to other types of detectors. The generaly low p-value across all the pair-wise comparisons also illustrate the steadiness of our key finds in the main text.

## M  LIMITATIONS AND FUTURE WORK

While **CodeMirage** represents a significant step toward a more comprehensive evaluation of AI-generated code detectors, several limitations remain and could be addressed in future work.

First, though **CodeMirage** includes a broad set of programming languages, LLMs, and detectors, it does not exhaustively cover all possibilities. Additional languages, particularly those less commonly used in mainstream software development but still important in specific domains, remain unexplored. Similarly, many emerging LLMs and detection techniques are not included in the current benchmark. Future work could expand **CodeMirage** to incorporate these newly emerged models and underrepresented languages, enabling broader and more inclusive evaluations.

Second, **CodeMirage** focuses primarily on document-level detection. With the rapid advancement and increasing adoption of sophisticated coding assistants, extending AI-generated code detection to multi-file or project-level contexts represents an important and meaningful direction. While our benchmark does not yet capture this dimension, we highlight it as a valuable direction for future exploration.

Despite these limitations, **CodeMirage** advances the field by offering a more comprehensive and realistic evaluation benchmark compared to prior work (Suh et al., 2025; Pan et al., 2024; Demirok & Kutlu, 2024; Pham et al., 2024; Orel et al., 2025; Xu et al., 2024b). We believe the insights obtained and evaluation platform established by **CodeMirage** will serve as a strong foundation for developing more robust and generalizable AI-generated code detectors.

## N  USE OF LARGE LANGUAGE MODELS (LLMS)

We employed LLMs only for copyediting (grammar and style). They did not contribute to the research design, analyses, writing of technical content, or interpretation of results. While our benchmark includes code generated by LLMs as test data, LLMs were not used to write or structure the paper. All scientific content and conclusions are by the authors.

