# OpenReview forum: "CodeMirage: Stress-Testing AI-Generated Code Detectors Against Production-Level LLMs"
_ICLR.cc/2026/Conference — ICLR 2026 Conference Withdrawn Submission_

### Official Review · Reviewer_Q3Zh · 2025-10-25

**Soundness:** 2
**Presentation:** 3
**Contribution:** 2
**Rating:** 4
**Confidence:** 4

**Summary:**

## Paper Summary

This paper introduces CodeMirage, a large-scale benchmark designed to evaluate and stress-test AI-generated code detectors under realistic, multilingual, and adversarial conditions. It addresses limitations in prior benchmarks that (1) focused on few programming languages, (2) relied on non–production-level LLMs, and (3) lacked adversarial scenarios such as paraphrasing. CodeMirage includes 210K samples across 10 programming languages, derived from human-written code (CodeParrot dataset), AI-generated code from 10 commercial LLMs, and AI-perturbed (paraphrased/adversarial) variants. The authors evaluate 10 representative detectors covering four paradigms (zero-shot, embedding-based, fine-tuned, and pretrained-LLM-with-classifier) under four configurations — in-distribution, out-of-distribution (cross-model/language), adversarial, and hybrid (OOD+adversarial).

**Strengths:**

## Strengths

1. Comprehensive Benchmark Coverage – CodeMirage spans 10 major programming languages and includes 10 diverse production-level LLMs (e.g., GPT-4o, Claude, Gemini, DeepSeek, Qwen), significantly improving realism over prior datasets.

2. Valuable Insights – The analysis (e.g., fine-tuning overfitting vs. zero-shot robustness) offers practical implications for deploying detectors in real-world environments.

3. Well-Structured and Readable – The paper is clearly written, well-organized, and presents technical content in an accessible and logical manner.

**Weaknesses:**

## Weaknesses

1. Limited Novelty – The novelty of this work appears modest. Comparing with related work in Table 1, the contribution primarily lies in integrating existing dataset design principles (multi-language coverage, adversarial perturbation, multi-model generation) rather than proposing a fundamentally new methodology. If the main contribution is the benchmark construction, similar ideas have already been explored in works such as CoDet-M4 (Orel et al., 2025) and LLMGCode (Xu et al., 2024b). If the contribution lies in evaluating more baselines, the paper does not introduce new detection methods or theoretical insights, and thus the advancement is mostly empirical rather than conceptual.

2. Motivation Needs Stronger Justification – The motivation for document-level detection remains unclear. The paper should better articulate why evaluating detectors on entire files (instead of function- or snippet-level, as in prior work) is necessary or more realistic. For example, are detectors expected to operate on full repositories in deployment? Or is document-level detection shown to capture distributional cues that function-level detection misses? Without such justification, the motivation appears weak.

3. Missing Implementation Details – Several approach details are under-specified. For AI-code perturbation, the six transformation types are only briefly mentioned, but not defined. Likewise, the implementation details for Multi-Round Paraphrasing, DeepWordBug, and AST-based Perturbation are omitted in the main text and deferred to appendices, making reproducibility difficult. A short description in the main section would improve clarity.

4. Lack of Fine-Grained Analysis –
   First, there is no detailed discussion on which specific LLMs produce code that is easier or harder to detect — although Figure 3 hints at this, the insight is not explicitly analyzed.

   Second, the evaluation setup seems largely model-agnostic, raising the question of how it leverages CodeMirage’s unique dataset properties. Since most perturbation techniques (e.g., paraphrasing, AST rewriting) are reused from prior work, the connection between the dataset design and the evaluation outcomes is unclear.

   Third, no dataset-specific evaluation (e.g., per-language or per-perturbation difficulty analysis) is provided to highlight what CodeMirage uniquely contributes beyond existing benchmarks.

5. Lack of Broader Impact or Future Vision – The paper does not clearly discuss how CodeMirage will influence future research or deployment. For example, will it enable detector training, robustness certification, or standardized evaluation protocols? Without articulating such a broader vision, the impact of CodeMirage may remain limited to a one-time empirical study rather than a lasting benchmark standard.

**Questions:**

1. What is the six transformation types in the adversarial perturbation generation?

2. If you conduct the same experiments on the dataset in existing work, will you get the same conclusion?

3. What is the motivation for document level detection?

---

### Official Review · Reviewer_CdYP · 2025-11-01

**Soundness:** 3
**Presentation:** 3
**Contribution:** 2
**Rating:** 6
**Confidence:** 3

**Summary:**

This paper introduces CodeMirage, a comprehensive benchmark for evaluating AI-generated code detectors under realistic and adversarial conditions. CodeMirage comprises approximately 210,000 code samples across 10 programming languages, including human-written code from GitHub and AI-generated/perturbed variants produced by 10 state-of-the-art production-level LLMs (including reasoning models like DeepSeek-R1, GPT-o3-mini, and Gemini-2.0-Flash-Thinking). The authors design six progressively challenging evaluation tasks across four configurations: in-distribution testing, out-of-distribution (cross-model and cross-language) testing, adversarial perturbation (primarily paraphrasing), and hybrid testing combining OOD shifts with adversarial attacks. Through extensive experiments with 10 representative detectors spanning four methodological paradigms (zero-shot, embedding-based, fine-tuning-based, and pretrained-LLM with downstream classifiers), the paper reveals several key findings.

**Strengths:**

+ The paper constructs a comprehensive and realistic benchmark that includes 10 programming languages and apply varied production-level LLMs to rigorously evaluate LLM-generated code detectors
+ The benchmark includes varied tasks to evaluate LLM-code detecotrs at different difficulty levels and from multiple perspectives
+ The paper presents a very comprehensive evaluation over existing LLM-code detectors and draw useful conclusions as clear takeaways.

**Weaknesses:**

- **Outdated code snippets for evaluation**. To avoid contamination with AI-generated code, the authors use the CodeParrot GitHub-Code-Clean dataset collected in May 2022, before the widespread deployment of modern code-generating LLMs. However, this reliance on pre-2022 code may introduce distributional shifts that affect the validity of findings, as coding practices, library usage patterns, and programming paradigms have evolved significantly since then. This setting raises concerns about whether the conclusions drawn from this benchmark generalize to contemporary code written today and in the future, particularly given that modern developers are evolving their coding behaviors by collaborating with advanced libraries and AI tools, potentially adopting significantly different patterns to detect unwanted AI usage or plagiarism.

- **Existing detectors seem to perform well for in-distribution data, challenging the difficulty and practical value of the benchmark**. As shown in Figures 3 and 9, many detectors achieve high F1 scores (often >0.85-0.95) in the in-distribution setting, with fine-tuned methods like GPTSniffer and CodeT5+ performing very well across languages and LLMs. This suggests that the in-distribution detection task may be too easy when training and test data share the same generator and language, potentially limiting the benchmark's ability to differentiate detector capabilities in this most basic scenario.

- **(Overly) aggressive filtering for potential memorized, AI-generated code**. The authors apply a conservative BLEU < 0.5 threshold to filter out potentially memorized AI-generated code, which may unintentionally introduce the distribution divergence between human-written and AI-generated code that can be easily captured by existing detectors. This aggressive filtering may artificially inflate the apparent difference required of AI-generated code in the benchmark, potentially biasing the dataset toward only highly divergent AI outputs while excluding realistic scenarios where AI models appropriately generate conventional solutions that naturally resemble human code patterns.

**Questions:**

- Could the authors conduct some analysis or provide some conceptual discussion to assess whether the temporal gap that includes only pre-2022 code introduces systematic biases for evaluating existing detectors with most recent code?

- Could the authors explain whether the low BLEU score filtering artificially introduces distribution divergence to make it easier for detectors to tell AI-generated code from human-written ones?

---

### Official Review · Reviewer_4SRw · 2025-11-07

**Soundness:** 2
**Presentation:** 4
**Contribution:** 2
**Rating:** 2
**Confidence:** 4

**Summary:**

This work explores and investigates the important task of code testing. It includes considerations of key data domains such as programming language types and challenges, and generates a large volume of test data. Additionally, this work leverages models as assistants to enhance the efficiency of data generation, making improvements and supplements to traditional approaches.

**Strengths:**

1: This paper conducts testing across a wide range of programming languages, effectively addressing the limitations of existing works that predominantly focus on mainstream backend programming languages such as C and Python.
2: The work considers the distribution of objective factors such as challenge levels and scene types, thereby providing a comprehensive data framework.
3: Through extensive experimentation, a large number of models are evaluated, leading to eight key findings.

**Weaknesses:**

1: The scale of the dataset remains a weak point, especially in the context of ten programming languages and a large number of LLM-assisted sample generation. There are concerns about whether the current dataset size can adequately support stress testing in the complex scenario of code generation.
2: There is insufficient discussion on how this work differentiates itself from existing studies, such as the peer-reviewed and published work Droid: A Resource Suite for AI-Generated Code Detection, and potentially other relevant works. These studies demonstrate significant overlap with the content of this work and are based on more comprehensive efforts, which undermines the contributions presented in this paper.
3: The use of LLMs for data generation is not a novel technique. Rather than focusing on the increased output of test data through LLMs, greater attention should be paid to ensuring the quality and diversity of synthesized test data, particularly for code—an inherently complex data structure. This aspect warrants further exploration.
4: There are minor issues in the paper’s presentation. While the number of categories and hierarchy are indeed essential considerations in dataset construction, the rationale and comprehensiveness of the classification scheme deserve more detailed discussion. Additionally, these categories lack a macro-level, systematic visual representation. The eight key findings in the experiments appear overly trivial and lack robust justification and deeper analysis. These issues cause the paper's intended message to be muddled and lacking in impact.

**Questions:**

1: Could the authors elaborate on the potential scalability of this work? For instance, how could the dataset size be increased through upgrades to the LLM or enhancements in resource allocation?
2: Have the authors considered the positive impacts of evaluation in this context? For example, how could this work help mitigate specific shortcomings in code generation tasks?
3: The authors should provide clearer visual representations and deeper explanations of the motivation behind the hierarchical classifications across different dimensions. Without such clarification, it would be difficult to accurately assess the true value of these figures.
4: Is it possible to present a more objective comparison of the advantages of this work relative to other recent, relevant studies?

---

### Note · Authors · 2026-01-06

I have read and agree with the venue's withdrawal policy on behalf of myself and my co-authors.